# Is Markerless More or Less? Comparing a Smartphone Computer Vision Method for Equine Lameness Assessment to Multi-Camera Motion Capture

**DOI:** 10.3390/ani13030390

**Published:** 2023-01-24

**Authors:** Felix Järemo Lawin, Anna Byström, Christoffer Roepstorff, Marie Rhodin, Mattias Almlöf, Mudith Silva, Pia Haubro Andersen, Hedvig Kjellström, Elin Hernlund

**Affiliations:** 1Sleip AI, Birger Jarlsgatan 58, 11426 Stockholm, Sweden; 2Department of Anatomy, Physiology and Biochemistry, Swedish University of Agricultural Sciences, 75007 Uppsala, Sweden; 3KTH Royal Institute of Technology, Division of Robotics, Perception and Learning, 10044 Stockholm, Sweden

**Keywords:** monocular motion analysis, objective lameness assessment, equine orthopaedics, animal pose estimation, optical motion capture

## Abstract

**Simple Summary:**

Lameness, an alteration of the gait due to pain or dysfunction of the locomotor system, is the most common disease symptom in horses. Yet, it is difficult for veterinarians to correctly assess by visual inspection. Objective tools that can aid clinical decision making and provide early disease detection through sensitive lameness measurements are needed. In this study, we describe how an AI-powered measurement tool on a smartphone can detect lameness in horses without the need to mount equipment on the horse. We compare it to a state-of-the-art multi-camera motion capture system by simultaneous, synchronised recordings from both systems. The mean difference between the systems’ output of lameness metrics was below 2.2 mm. Therefore, we conclude that the smartphone measurement tool can detect lameness at relevant levels with easy-of-use for the veterinarian.

**Abstract:**

Computer vision is a subcategory of artificial intelligence focused on extraction of information from images and video. It provides a compelling new means for objective orthopaedic gait assessment in horses using accessible hardware, such as a smartphone, for markerless motion analysis. This study aimed to explore the lameness assessment capacity of a smartphone single camera (SC) markerless computer vision application by comparing measurements of the vertical motion of the head and pelvis to an optical motion capture multi-camera (MC) system using skin attached reflective markers. Twenty-five horses were recorded with a smartphone (60 Hz) and a 13 camera MC-system (200 Hz) while trotting two times back and forth on a 30 m runway. The smartphone video was processed using artificial neural networks detecting the horse’s direction, action and motion of body segments. After filtering, the vertical displacement curves from the head and pelvis were synchronised between systems using cross-correlation. This rendered 655 and 404 matching stride segmented curves for the head and pelvis respectively. From the stride segmented vertical displacement signals, differences between the two minima (MinDiff) and the two maxima (MaxDiff) respectively per stride were compared between the systems. Trial mean difference between systems was 2.2 mm (range 0.0–8.7 mm) for head and 2.2 mm (range 0.0–6.5 mm) for pelvis. Within-trial standard deviations ranged between 3.1–28.1 mm for MC and between 3.6–26.2 mm for SC. The ease of use and good agreement with MC indicate that the SC application is a promising tool for detecting clinically relevant levels of asymmetry in horses, enabling frequent and convenient gait monitoring over time.

## 1. Introduction

Objective measurement of a horse’s motion at the trot has become an important part of the diagnostic procedures performed during clinical lameness investigation. These measurements, which have been used in clinical practice for more than a decade, trace the vertical displacement of axial body segments: the head, the pelvis and sometimes the withers. Using reflective markers or inertial sensors attached to a point on each body segment, a time series signal is generated. In trot, the vertical displacement signal takes the shape of a sinusoidal double wave from each stride and it is the position of the two peaks and valleys of this signal which are used for lameness analysis.

The degree of asymmetry in the vertical displacement signal i.e., the difference between the two peaks and valleys respectively is known to indicate asymmetric loading of the left versus right limb during the midstance and the push-off phases of the stride [1,2,3]. Measurements of these asymmetries provide the veterinarian with high-resolution data that help overcome the limited time resolution of the human visual system [4,5]. These objective data seem crucial for quality control of the clinical procedure, since subjective lameness assessment has been shown to have moderate to low agreement between veterinarians [6,7] and is affected by expectation bias [8]. The metrics derived from objective motion analysis show high sensitivity for single-limb lameness, acting as early indicators to detect asymmetric loading of the limbs [9]. However, the specificity for lameness on a population level is less clear. Motion asymmetries are commonly observed in cross-sectional studies of different horse populations, such as Warmblood riding horses [10,11], Thoroughbred race horses [12], working Polo horses [13], elite eventing horses [14], endurance horses [15] and young Standardbred trotters [16], with a prevalence ranging between 50–90 percent. Although these asymmetries are of the same magnitude as in horses investigated for lameness in a clinical setting [17], it is currently unknown if these asymmetry levels indicate that a large proportion of horses in training are lame or if the asymmetries can be explained by other factors, such as laterality. A key approach to the further investigation of this issue is to perform longitudinal monitoring of individual horses over time. For this to be possible, a reliable, ease of use and low-cost measuring system is required.

Several motion analysis systems have been developed for clinical use based on inertial measurement units (IMUs) [18,19,20]. Also available is a multi-camera marker-based motion capture (MC) system [21]. This MC-system, is considered to be the gold standard for measuring body segment movement for kinematic gait analysis [22]. It relies on reflective markers attached to a horse’s body. These markers are detected and tracked by a set of cameras that are geometrically calibrated and temporally synchronized. By using the synchronized tracks of the marker positions in the camera images, MC-systems reconstruct the 3D coordinates using multi-view triangulation. It has been shown that under favourable conditions, the accuracy of the computed estimates of the 3D positions over time is less than a millimetre for the MC-systems [23]. However, placing markers on the horse are resource and time-consuming in a clinical situation, and the equipment is a substantial financial investment for a veterinary practice. This impedes the system’s large-scale clinical and scientific use.

During the previous decade, the field of computer vision was revolutionized by methods based on deep neural networks [24,25,26]. These networks are computer algorithms that consist of multi-layered (referred to as deep) compositions of parametric functions that can be trained on large datasets to perform classification and regression tasks. Deep learning has demonstrated increased robustness to differing scenarios, light conditions, and noise levels compared to traditional computer vision methods. Estimation of poses of the human body from images has been enabled by deep learning and as a result of this development, it has become possible to perform motion analysis from video, e.g., from a smartphone camera [27]. Recent works on horse lameness classification [28,29] have demonstrated a progressive movement towards the application of computer vision and deep learning within objective motion analysis. However, a binary disease classifier of “lame” versus “not lame” is a difficult approach for clinical use, given that lameness is often not a binary state and that the deep learning algorithms act in a black box manner, rendering distrust from medical professionals [30]. Instead, providing a clinician with computer vision derived metrics to support medical decision making is a more implementable approach. But until now, the methodological accuracy of deep neural networks for quantification of clinically used lameness metrics has not been investigated.

In this work, we validate a new single-camera markerless (SC) system designed for equine lameness assessment which uses images from a smartphone camera video stream. To achieve robust detection and tracking, the system employs a series of neural networks. These networks were trained to detect and track the pelvis, head and hooves in video streams of trotting horses. The system also detects the trotting direction of the horse, away from or towards the camera, to determine which parts of the horse are visible for measuring. The network designs were inspired by previously proposed methods for object detection [28,31] and segmentation [32,33]. Unlike the MC-system, the deep neural networks of the SC-system do not require that markers are placed on the horse. Instead, the networks learn to detect the points of interest on the horse’s body visible in the images through training on large datasets.

The specific aim of the study was to compare this new markerless smartphone system to a state-of-the-art multi-camera marker-based system with respect to waveform similarity of the derived vertical displacement signals and the limits of agreement for their extracted lameness metrics.

## 2. Materials and Methods

### 2.1. Study Protocol

Twenty-five horses were recorded as they underwent motion analysis at the orthopedic gait laboratory situated in the Equine Clinic of the University Animal Hospital in Uppsala, Sweden. The recordings were performed simultaneously with a multi-camera marker-based motion capture system and a single smartphone camera markerless system. The experimental setup is illustrated in Figure 1. The study subjects were a convenience sample selected from the horses visiting the clinic during the 10-day data collection period, without any exclusion criteria. The horses were of different breeds, sizes (range 128 to 180 cm to the withers) and colours (black, bay, chestnut, grey). All owners gave their written informed consent to participate. The study did not in any way alter the clinical procedures or add physical manipulation of the animals. Hence, no ethical approval was required according to the national animal ethics legislation.

### 2.2. Data Collection

During data collection, each horse was guided by a handler (the owner, or a researcher running at the horse’s left side) to trot at least two times back and forth on a 30 m concrete runway in a corridor. The horses were jogged at the handler’s preferred speed.

We employed a MC-system with 13 cameras (Qualisys AB, Motion Capture Systems). The MC cameras were placed ≈4 m over the ground and in a manner such that the union of the field of view of each camera covered as much of the runway as possible while still maintaining sufficient overlap between neighboring cameras to perform tracking and triangulation of the marker positions. The recording rate was set to 200 Hz. We attached spherical reflective markers in the median plane over the poll on the horse’s head and between the tubera sacrale of the pelvis, allowing the MC-system to detect and track the markers over time. Additional markers were placed, but not used in this study (see Figure 2).

The smartphone (iPhone12 Pro Max) was placed ≈1.6 m above the ground on a tripod facing the direction of the horse trot recording 4k video (2160×3840 pixels) at 60 Hz. The video streams recorded were input to the SC-system. Example frames from the smartphone camera recordings are shown in Figure 2.

For each recording, the MC-system and the smartphone camera were triggered at approximately the same time such that the data streams from both systems would cover the same sequence of events.

### 2.3. Signal Extraction

The MC-system software (Qualisys Track Manager—QTM, Qualisys AB) automatically tracked the reflective markers and generated 3D coordinates corresponding to the positions of the markers in each frame. The 3D marker coordinates from QTM were exported to .mat files (MATLAB). From these coordinates, vertical displacements were extracted for each frame, one for the head marker and one for the pelvis marker. This resulted in two vertical displacement signals (VDS), $y_{mc}^{head}\left(t\right)$ and $y_{mc}^{pelvis}\left(t\right)$ for head and pelvis respectively, where *t* is time in seconds.

For the SC-system, deep neural networks were applied (software of Sleip AI AB) on the input video stream from the smartphone camera. The deep neural networks were trained to output the pixel coordinates of horse body parts for each frame of the video. The training material for the deep neural networks contained horses of many different coat colours and varying conformation, but none had physical markers attached to the skin. Head ($y_{sc}^{head}\left(t\right)$) and pelvis ($y_{sc}^{pelvis}\left(t\right)$) VDS was calculated from the pixel coordinates. Additionally, the VDS of all four hooves were extracted from the SC-system for stride splitting purposes (see Section 2.4). Both the MC-system and the SC-system data were further processed using custom written python scripts.

Note that the pelvis was visible to the SC-system only when the horse was trotting away from the camera, while the head was mostly visible in both directions. Consequently, SC generally produced data from a higher number of strides with head tracking than strides with pelvic tracking. The following analyses only included strides with data matched from both systems for the body segment in question (head and pelvis). Horses were removed from the dataset where less than 10 matching strides were available for either head or pelvis since we deemed that insufficient for statistical relevance.

### 2.4. Stride Split and Signal Filtering

The recorded data contained noise, due to measurement errors and because the horses seldom trot in a consistent manner throughout a trot-up. This noise was present for both the MC-system and the SC-system data. Thus, the VDS had to be band-pass filtered in order to remove the noise without affecting the frequency content of the signal that related to movement asymmetries and lameness [34].

In order to perform the VDS filtering described in [34], the within horse mean stride frequency of a measurement was needed. This was estimated by extracting strides from the hoof VDS of the SC-system. Firstly, we performed a pre band-pass filtering of the hoof VDS to remove trends and high-frequency noise. Specifically, a 7th order Butterworth digital filter with a lower bound cut-off frequency of 0.6 Hz and an upper bound cut-off frequency of 2.2 Hz. This allowed us to determine in which time intervals the left hoof was above the right hoof and vice versa, ultimately enabling the classification of left and right strides.

Next, the lengths of the intervals were used to compute the stride frequency, which in turn was used to set the bounds of the 10th order Butterworth band-pass filter applied to the VDS of the pelvis and head. Specifically, we set the lower bound to 0.75 times the stride frequency, to not alter frequency content related to the movement asymmetry [34]. Similarly, the upper bound was set to 2.42 times the stride frequency to not attenuate the frequency content related to the symmetric movement, while omitting higher frequency content and noise.

### 2.5. Signal Synchronization

Since the MC and SC recordings were triggered manually, the extracted signals were not adequately synchronized in time. To synchronize the vertical displacement signals we computed cross-correlations to find the relative time shift $t_{shift}$ that solved the following maximization problem:(1)$$t_{shift}=\underset{t}{arg max}\left(\left(y_{mc}^{pelvis}🟉y_{sc}^{pelvis}\right)\left(t\right)+\left(y_{mc}^{head}🟉y_{sc}^{head}\right)\left(t\right)\right)\;.$$

Here 🟉 denotes the correlation operator. The signals were band-pass filtered according to Section 2.4 before synchronization.

### 2.6. Asymmetry Quantification

In this section, we introduce a number of definitions that we use in the remaining parts of the paper. First, we define a stride segment as a section of the VDS corresponding to a time interval of a full stride. To extract the stride segment we utilize the extreme values (see the illustration in Figure 3). Tracing of the vertical displacement of the horse’s head or pelvis while it trots yields a sine-shaped signal as depicted in Figure 3. The valleys (local minima) and peaks (local maxima) of the signal are associated with the vertical forces generated during impact (the more force, the lower the valley position) and the relative timing of horizontal and vertical forces during the propulsive phase of the stride respectively (less push-off rendering a lower peak position). Differences between consecutive peaks or valleys can be quantified into an asymmetry index [22], which can be used as an indicator of lameness severity. Depending on the measurement technique and due to variation in horse size, these values may need to be normalised to be comparable [35,36]. Therefore, normalisation of these values to the range of motion (R) is used in the SC-system to obtain values that are more independent of horse size.

#### 2.6.1. Extraction of Valleys and Peaks

To find the extreme values within each stride segment (the VDS from one stride), consecutive data points were compared to find points at which the derivative of the VDS was zero. Let y(t) be the VDS value at time *t*. We defined the peaks $p_{i}=y\left(t_{p_{i}}\right),i=1,2,\ldots$, as the local maximum values and the valleys $v_{j}=y\left(t_{v_{j}}\right),j=1,2,\ldots$, as the local minimum values. Further, we assumed that $t_{p_{i}}<t_{p_{i+1}}$ and $t_{v_{j}}<t_{v_{j+1}}$. We extracted a sequence of consecutive peaks and valleys $p_{i},p_{i+1},v_{j},v_{j+1}$ such that $t_{p_{i}}<t_{v_{j}}<t_{p_{i+1}}<t_{v_{j+1}}<t_{p_{i+2}}$.

In horses showing moderate to severe lameness, the changes in motion asymmetry can cause extreme asymmetries in y(t). In these cases, local extreme values might be canceled out, interrupting the assumed stride pattern of two peaks and two valleys in sequence. Instead, a single peak or valley signal pattern occurs. To handle this, we implemented a robust extreme value extraction method. The reasoning behind this method is that y(t) contains two dominant harmonics [1]. The first harmonic corresponds to the stride frequency, thus contributing to the asymmetry of the signal. The second harmonic corresponds to twice the stride frequency and should dominate y(t) if the horse is healthy. In our approach, we extracted extreme values based on the curve shape of the second harmonic. We first removed the asymmetric component of y(t) by performing high-pass filtering with a frequency bound higher than the first harmonic. Next, we employed local extreme value extraction on the high-pass filtered signal. Finally, we refined the selection by selecting the extreme values from the original y(t) within 50 ms of the estimated value. As a result, we were able to estimate normalised peak/valley differences at any degree of lameness.

#### 2.6.2. Normalised Differences for Valleys and Peaks

From the extracted stride peaks we computed local extreme value differences per stride *i*, consisting of the following scalars,
(2)$$\begin{array}{cc} \; & {\mathrm{MinDiff}}_{i}=v_{i+1}-v_{i} \end{array}$$
(3)$$\begin{array}{cc} \; & {\mathrm{MaxDiff}}_{i}=p_{i}-p_{i+1}\;. \end{array}$$

These values provide information about the asymmetries between the right and left leg impact and push-off. However, since different horses have different amplitudes in their vertical displacement trajectories, these values are not comparable. In this work, we instead use the normalised extreme value differences (NEVd), which show more independence from the scale of the vertical displacement. The NEVd-values were computed using the following operations,
(4)$$\begin{array}{cc} \; & V_{i}=\frac{{\mathrm{MinDiff}}_{i}}{R_{i}} \end{array}$$
(5)$$\begin{array}{cc} \; & P_{i}=\frac{{\mathrm{MaxDiff}}_{i}}{R_{i}}\\ \; & \mathrm{where}\;R_{i}=\max\left(p_{i},p_{i+1}\right)-\min\left(v_{i},v_{i+1}\right)\;. \end{array}$$

Here, normalization was performed by division by the range-of-motion $R_{i}$. Thus, the NEVd-values measures asymmetry as a rate of $R_{i}$.

#### 2.6.3. Outlier Removal

To remove occasional strides with erroneous measurements from the analysis, we performed a series of outlier removal steps.

While the noise in the vertical displacement signals is partly suppressed by band-pass filtering, the signal quality becomes inadequate for asymmetry analysis when the noise is too prevalent. Therefore, we first removed strides where the stride segments contained a substantial amount of high-frequency noise. A stride segment was deemed to contain too much high frequency noise when the majority of the frequency content amplitudes were found above 10 Hz in at least a quarter of a stride interval.

As a second step, we performed a linear discriminant analysis (LDA). In addition to ${\mathrm{MinDiff}}_{i}$ and ${\mathrm{MaxDiff}}_{i}$ from Equation (Equation 4), we used the peak valley differences $p_{i}-v_{i}$ and $p_{i+1}-v_{i+1}$ to represent each stride as features. We then removed NEVd-values from the analysis that corresponded to strides that were considered outliers in the LDA.

### 2.7. System Comparisons

We compared the MC and SC systems on the data set of trotting horses described in Section 2.2. For each recorded sample, we used MC and SC to generate vertical displacement signals. From these, we extracted and compared stride segments and NEVd-values between the two systems. The following sections detail the implementation and setup of the comparison.

#### 2.7.1. Comparison Metrics

To compare the extracted asymmetry indices between the MC and SC systems the following deviations were calculated from the synchronized NEVd-vales,
(6)$$\begin{array}{cc} \; & \Delta V^{i}={\mathrm{MinDiff}}_{i}^{sc}-{\mathrm{MinDiff}}_{i}^{mc} \end{array}$$
(7)$$\begin{array}{cc} \; & \Delta P^{i}={\mathrm{MaxDiff}}_{i}^{sc}-{\mathrm{MaxDiff}}_{i}^{mc}\;. \end{array}$$

To recover an estimate of geometric deviation, we multiplied $\Delta V^{i}$ and $\Delta P^{i}$ with the range of motion $R_{i,mc}$ computed from the NEVd-values of the MC signal.

In practice, lameness indication is deduced from the trial mean of the NEVd-values $\bar{V}=1/N\sum_{i}^{N}{\mathrm{MinDiff}}_{i}$ and $\bar{P}=1/N\sum_{i}^{N}{\mathrm{MaxDiff}}_{i}$, where *N* is the number of strides in the trial after the outlier removal in Section 2.6.3. We computed the trial mean deviations as,
(8)$$\begin{array}{cc} \; & \Delta \bar{V}={\bar{V}}^{sc}-{\bar{V}}^{mc} \end{array}$$
(9)$$\begin{array}{cc} \; & \Delta \bar{P}={\bar{P}}^{sc}-{\bar{P}}^{mc}\;. \end{array}$$

We further scaled $\Delta \bar{V}$ and $\Delta \bar{P}$ with the mean range of motion ${\bar{R}}_{mc}=1/N\sum_{i}^{N}R_{i,mc}$ to estimate the geometric deviation.

#### 2.7.2. Statistical Analysis

Bland-Altman analysis [37] was used to evaluate the statistical agreement between the MC and SC-systems for head and pelvis NEVd-values. The Bland-Altman analysis was subdivided into deviations for MinDiff and MaxDiff and was performed both on trial and stride level.

In addition to the comparison of the NEVd-values, we compared the shapes of the band pass filtered vertical displacement signals using the root mean square deviations (RMSD),
(10)$$\begin{array}{cc} \; & RMSD=\sqrt{\frac{\sum_{m=1}^{M_{i}}{\left(R_{i,sc}^{mc}\cdot y_{sc}\left(t_{m,sc}^{i}\right)-y_{mc}\left(t_{m,mc}^{i}\right)\right)}^{2}}{M_{i}}}\\ \; & \mathrm{where}\;R_{i,sc}^{mc}=\frac{R_{i,mc}}{R_{i,sc}}\;. \end{array}$$

Here, *M* = 23 is the number of trials, i.e number of horses in the dataset. $t_{m,sc}^{i}$ and $t_{m,mc}^{i}$ are equally spaced points in time for sampling the vertical displacement signals corresponding to the *i*th synchronized stride, i.e $t_{1,sc}^{i}=p_{i,sc}$, $t_{M,sc}^{i}=p_{i+2,sc}$, $t_{1,mc}^{i}=p_{i,mc}$ and $t_{M_{i},mc}^{i}=p_{i+2,mc}$. Note that we scaled the $y_{sc}$ samples to the geometric scale of $y_{mc}$. Further, since $y_{sc}$ and $y_{mc}$ have different frame rates, we aligned and re-sampled the signals with linear interpolation before applying (Equation 10).

## 3. Results

In this section, we outline the results from the experiments described in Section 2.7. In total, the results below were generated from 23 of the 25 horses in the initial data set, after the exclusion of two horses with less than 10 synchronized strides. From the included horses we extracted a total of 655 stride observations of the head motion and 404 stride observations of the pelvic motion. Descriptive statistics of the head measurements from the 23 trials can be found in Table 1 and the pelvis measurement in Table 2.

We split the comparisons into per-stride and per-trial comparisons. In the per-stride comparisons, we treated each stride as a sample and performed statistical analysis on the stride-based deviations $\Delta V_{i}$ and $\Delta P_{i}$. In the per-trial comparison, we treated each horse as a sample and performed statistical analysis on deviations computed from the mean NEVd values $\Delta {\bar{V}}_{i}$ and $\Delta {\bar{P}}_{i}$.

### 3.1. Per-Stride Comparisons

An example of the time-domain curves from the MC and SC for all strides in a recorded trial (horse 11) are displayed in Figure 4. The displayed stride segments of the two systems show high resemblance, resulting in similar conclusions on lameness diagnosis. We provide more examples from the experiment in Appendix A.

The Bland-Altman analysis for head and pelvis lameness metrics is illustrated in Figure 5 and Figure 6 respectively. The deviations are similar for both valley and peak differences and generally higher for head than for pelvis signals. Moreover, the correlations between the NEVd-values are high and the deviations are generally small, rarely exceeding 21 mm for the head signals and 14 mm for pelvis.

We further provide histograms over the deviations for head and pelvis in Figure 7. The histograms include both *V* and *P*-values. In addition, these plots show the empirical estimates of normal distributions computed from the stride samples. The distributions of the absolute values for stride mean residuals are presented in Figure 8.

Finally, we provide mean RMSD values in Table 3 to give an estimate of curve similarity. Note that RMSD, as computed in (Table 3), is sensitive to small time shifts. As the synchronization between signals $y_{sc}\left(t\right)$ and $y_{mc}\left(t\right)$ is approximate, the RMSD does not only reflect curve similarity but also errors in the synchronization.

### 3.2. Per-Trial Comparisons

In this section, we provide the results from the per-trial comparison between MC and SC. In Table 3 we present statistics over the entire dataset from the per-trial mean NEVd-values. In this case, we combine $\bar{V}$ and $\bar{P}$. Thus, each stride contributes with two deviation values. For these, we compute the overall absolute mean, maximum and minimum absolute deviations as,
(11)$$\begin{array}{cc} \; & \bar{D}=\frac{\sum_{m=1}^{M}|\Delta {\bar{V}}_{m}|+\sum_{m=1}^{M}\left|\Delta {\bar{P}}_{m}\right|}{2M} \end{array}$$
(12)$$\begin{array}{cc} \; & \max D=\max(\{|\Delta {\bar{V}}_{m}{\left|\right\}}_{m=1}^{M}\cup \{|\Delta {\bar{P}}_{m}{\left|\right\}}_{m=1}^{M}) \end{array}$$
(13)$$\begin{array}{cc} \; & \min D=\min(\{|\Delta {\bar{V}}_{m}{\left|\right\}}_{m=1}^{M}\cup \{|\Delta {\bar{P}}_{m}{\left|\right\}}_{m=1}^{M})\;, \end{array}$$
where M=23 is the number of trials, i.e number of horses in the dataset, and ∪ denotes the union of the sets of $\{|\Delta {\bar{V}}_{m}{\left|\right\}}_{m=1}^{M}$ and $\{|\Delta {\bar{P}}_{m}{\left|\right\}}_{m=1}^{M}$.

Similar to the per-stride comparison in Section 3.1, we use Bland-Altman plots [37] to inspect the statistical agreements. The plots for head and pelvis are shown in Figure 9 and Figure 10, respectively. Not unexpectedly, the deviations between systems are smaller for the per-trial mean NEVd-values than for the per-stride comparison in Section 3.1, rarely exceeding 6.4 ms. Similar to the per-stride comparison, the differences between $\bar{V}$ and $\bar{P}$ are small. However, the per-trial deviations show similar values for pelvis and head, which could be due to the fact that the two systems jointly observed more strides for the head signal than the pelvis.

In addition, we provide histograms over all the deviations (both $\bar{V}$ and $\bar{P}$ ) for head and pelvis in Figure 11. These plots also show the empirical estimates of normal distributions computed from the stride samples.

## 4. Discussion

In this work, we demonstrated that deep neural networks and computer vision can be applied to reliably perform orthopaedic gait analysis for horses when trotted in hand on the straight during lameness assessment. The recorded average per-trial errors of 2.17 mm (head) and 2.19 mm (pelvis) are well below the previously recorded between-measurement-variation, of 18 mm (head) and 6 mm (pelvis) of horses trotting in a MC-system with several repetitions over two consecutive days and a one-month follow-up [21]. The results thus indicate that the average per-trial errors of the SC-extracted variables compared to the MC-extracted variables were small enough to not be a major hindrance when used for objective lameness assessment.

From a clinical perspective, the ease-of-use of the SC-system studied has a clear benefit since it allows affordable, repeated observations of equine patients, which can help to understand the considerable between trial variations observed in gait asymmetry [21].

When comparing the results of this study to another validation study performed on an IMU-based system (compared to a MC-system), it was found that stride-by-stride limits of agreement for the pelvic variables were approximately twice the magnitude for the SC-system described in this work [20]. It has to be acknowledged that comparing results from different samples can be a confounding factor in this case, but it still presents a general indication of how a computer vision based solution may compare to an IMU-based solution. Unfortunately, limits of agreement for the IMU-system in the Bosch et al. study [20] versus our SC-system can not be compared for the head variables since these were not presented in the IMU-system validation study.

The reported accuracy of the SC-system could also be compared to a previous test-retest repeatability study of an IMU-system, where the 95% confidence interval was reported as approximately 6 mm for head asymmetries and 3 mm for pelvic asymmetries [38]. However, a later study comparing that IMU-system to a different IMU-system, also developed for detecting equine lameness, found that the limits of agreement between the two systems were in this same range [36]. Further, it was found that the system used by [38] consistently underestimated the amount of movement asymmetry compared to the other IMU-system, which had previously been shown to give values comparable to optical motion capture. It has been suggested that the confidence intervals reported by Keegan et al [38] should be adjusted to 8 and 4 mm [13] for the other IMU-system, and this is likely a suitable adjustment also for an MC-system.

There were two outlier trials with a recorded per-trial mean deviation of 8.71 mm (head) and 6.54 mm (pelvis). Notably, these outliers occurred for horses with large asymmetries (see horse 2 in Table 1 and horse 4 in Table 2), and did not change the sign of the calculated variables or the clinical interpretation of the gait data. Hence, these deviations would not confuse which limb was affected by the asymmetry. We hypothesized that these errors might be due to difficulties in detecting and tracking the head and pelvis when they were occluded e.g., the horse lowering its head and hiding it behind the trunk, or lifting the tail obscuring the pelvic region.

Another SC-system utilizing deep neural networks to detect and track trotting horses for the purpose of lameness assessment has been previously described [39]. Although their approach was similar to the method presented in this study, the authors did not quantify movement symmetry metrics directly. Instead, they focused on lame limb classification. It is also worthwhile noting that a SC-system has been developed and shown to be able to perform reliable gait analysis of human subjects [27]. The level of precision described is said to open up for several potential applications in human medicine.

Is markerless more or less? A dichotomous answer cannot be given here. There are technical drawbacks related to using computer vision techniques implemented in a SC-system aimed towards objective equine lameness assessment, lower accuracy and lack of 3D motion to name two. However, these drawbacks have to be weighed against the benefit of having a lightweight, portable and low-cost system available for data collection, that allows repeatable observations of the horse. Other systems, such as IMUs and MC are typically more expensive, are sometimes limited to laboratory environments and require more interaction in terms of placing markers and sensors on the horse. Inevitably, this leads to fewer measurements being done and it is well known that low sample sizes of horses are the standard in many equine biomechanics studies. This study has shown that a simple application on a smartphone can be a tool for flexible and reliable collection of kinematic asymmetry data from horses. By extension, this opens up opportunities for larger-scale biomechanics research studies in non-laboratory environments. Coupling this with current advances in machine learning, where computers efficiently learn from data to perform predictions, we suggest that SC can be used to accelerate our understanding of horse locomotion and horse welfare.

### Limitations

In the current study, all measurements were performed in the same clinical indoor environment on a limited number of horses (n = 23). The SC-system would likely be more challenged if there was a severe lack of light or if for example heavy rain obscured the visibility of the horse in the video. However, testing under such conditions was not within the scope of this study and would have been impossible to perform given that the state-of-the-art MC-system is a permanent indoor installation. The smartphone was placed on a tripod during the data collection, as handheld recordings would demand a stabilisation algorithm to be applied to the video. As such, further research is required to evaluate the SC-system under handheld conditions. Also, the length of the runway was 30 m, hence a greater distance between the camera and the horse has not been investigated. We did, however, analyse the error per stride index and did not find increasing deviations of the lameness metrics studied.

There are today several iPhone models with different camera specifications, such as image resolution and sampling rate. This study was limited to the use of an iPhone12 Pro Max where the resolution was set to 2160 × 3840 pixels and the frame rate to 60 Hz. Newer iPhone models come with even better camera specifications. Further research should investigate whether these models would improve the output from the SC-system even further.

The neural networks used in the SC-system were trained on image data of horses that did not have markers attached to the skin. Therefore, the skin-attached reflective markers used in this validation study could be suspected to partly obscure the anatomical region of interest and potentially present a problem to the SC-system. Visual inspection of the tracking from the SC-system, did however show very stable detection of the anatomical segments. This is confirmed by the comparison to the output of the MC-system.

This study only investigated vertical movement asymmetry of head and pelvis in a straight line. 3D motion comparisons would be of interest in the future, in order to provide a more detailed analysis of horse locomotion e.g., limb retraction and protraction angles studied from lateral view or on a circle. However, 3D lifting from a 2D image is an inherently complicated computer vision task where more advanced methods would be required, and so these kinematic variables were not investigated in this preliminary study.

## 5. Conclusions

We conclude that objective gait analysis for lameness assessment in horses can be reliably performed using a smartphone and computer vision analysis built on deep neural networks. The measurement deviation, when compared to a state-of-the-art motion capture system, is larger compared to IMU-based systems [20], but the error is clearly lower than observed levels of “between trial variation” from earlier studies [21]. The ease-of-use of the system makes repeated observations of a horse’s lameness more feasible, which can provide more objective data points for treatment evaluation.

## Figures and Tables

**Figure 1 animals-13-00390-f001:**
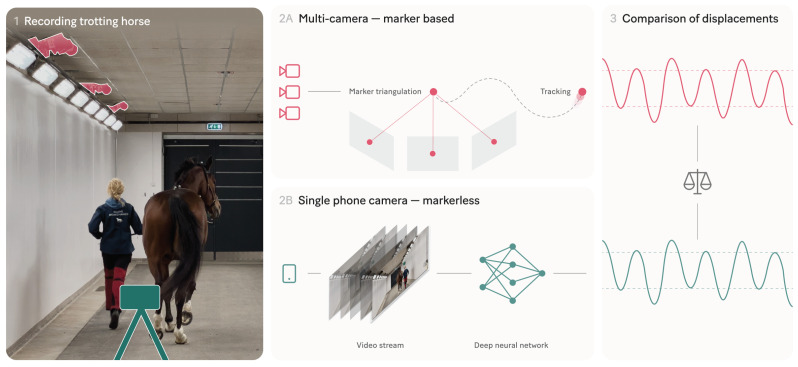
Illustration of the experimental setup. We recorded horses trotting back and forth in a corridor with a multi-camera system and a single smartphone camera. The multi-camera system detected and tracked reflective markers attached to the horse. Marker positions were triangulated into 3D coordinates from which vertical displacement curves were extracted. The single-camera system used deep neural networks to predict the vertical displacement curves of the head and pelvis from the images in the smartphone video stream. We then compared the displacement curves from the two systems using normalised peak and valley differences.

**Figure 2 animals-13-00390-f002:**
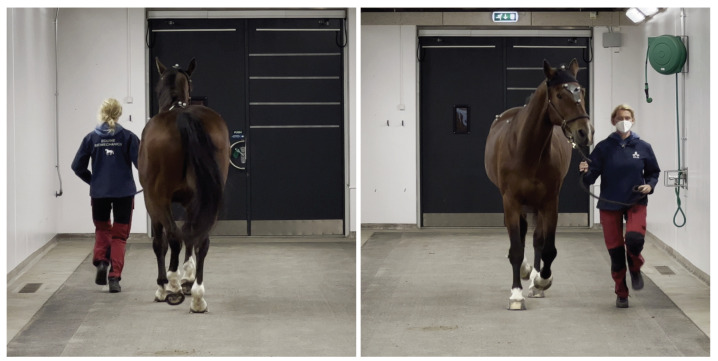
Example images from data set (horse 21) taken from the video recorded by the markerless single-camera system (SC). Attached to the horse’s skin by double adhesive tape are the spherical reflective markers used by the multi-camera marker-based system (MC) for tracking head and pelvic motion. The markers on the poll and between the tubera sacrale were used.

**Figure 3 animals-13-00390-f003:**
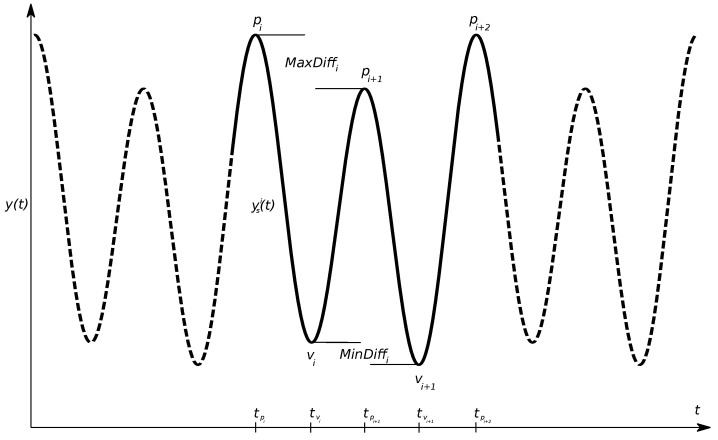
Example of the vertical displacement signal from head or pelvis, with local minima denoted with *v* for valley and local maxima with *p* for peak. Metrics for lameness quantification are calculated as the difference between the two minima (MinDiff) and the two maxima (MaxDiff) per stride. A stride segment $y_{s}^{i}\left(t\right)$ was defined as the section of y(t), starting at $t_{p_{i}}$ and ending at $t_{p_{i+2}}$.

**Figure 4 animals-13-00390-f004:**
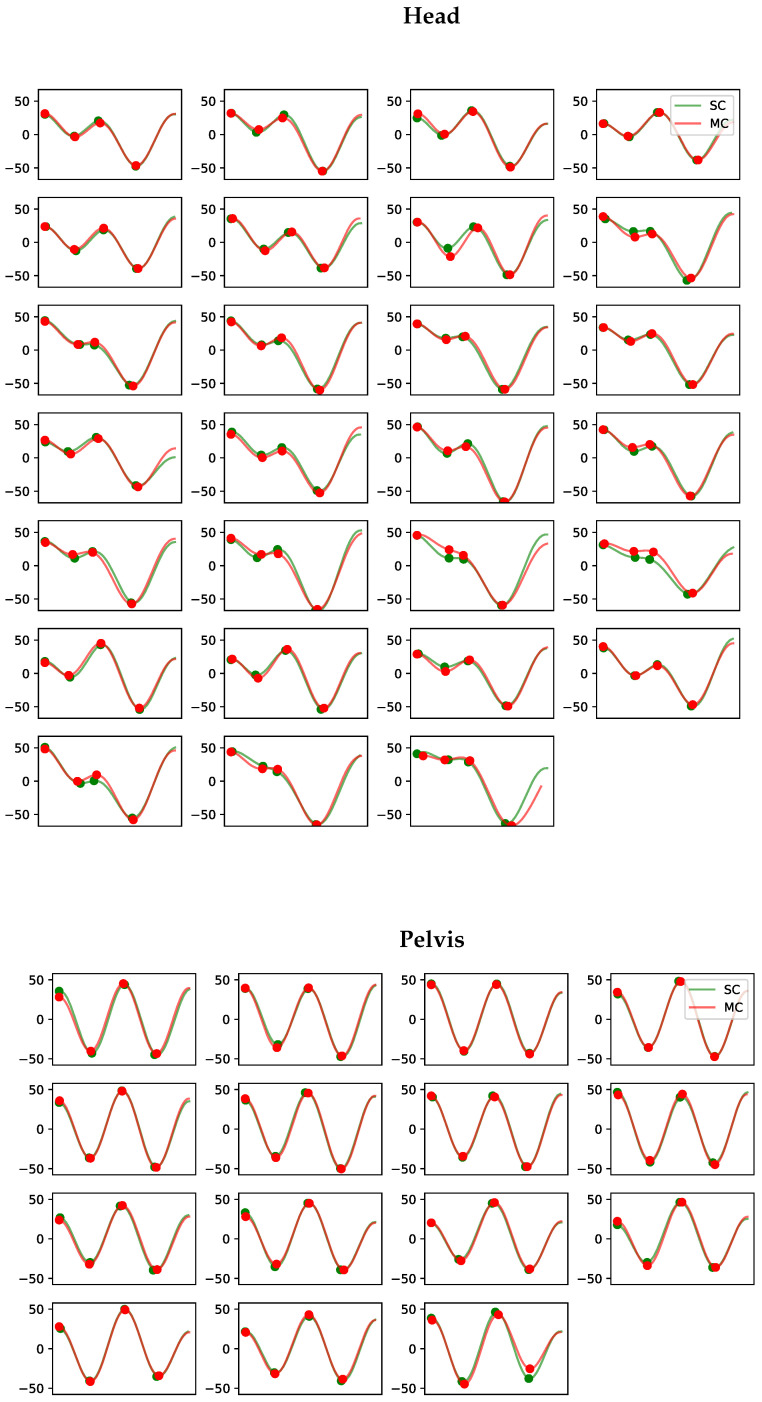
Example of the vertical displacement signal per stride for the head and pelvis for horse 11. Each subplot contains the matched stride segments for the markerless single-camera system (SC) in green and the multi-camera marker-based motion capture system (MC) in red. The y-axis shows the vertical displacement in millimetres. The four dots on each curve indicate the positions of the two peaks and two valleys extracted using the approach in Section 2.6.2. We observe that despite the high variability in curve shape between strides, there is a notable resemblance between the two systems.

**Figure 5 animals-13-00390-f005:**
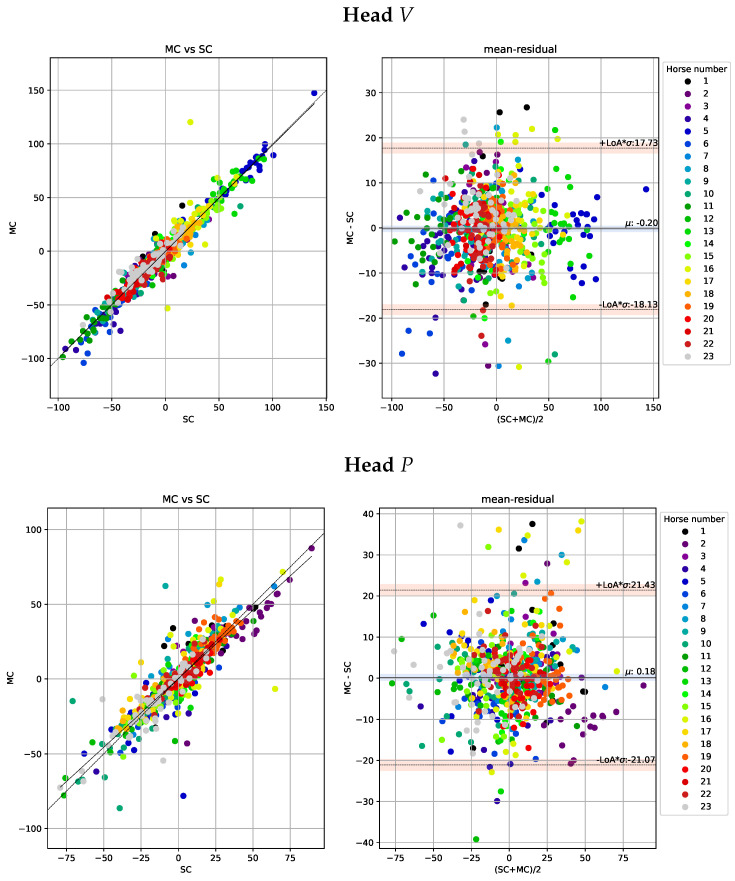
Scatter plots of the head metrics obtained per stride (n = 655) observed by the multi-camera marker-based motion capture system (MC) plotted against the observation by the single-camera markerless system (SC) are shown in the left sub-panels. Agreements between the systems, with limits of agreement (LoA) displayed as orange horizontal lines, are presented in the Bland-Altman plots in the right sub-panels. In the top row, we show the ΔV (MinDiff) and in the bottom row, we show ΔP (MaxDiff) defined in Section 2.7.1. For the purpose of visibility, we have fixed the range of the y-axis, causing a few samples with large residuals to be out of range. In the left plot, we have out of range residuals at 97.0 and −55.2, and in the right plot out of range residuals are at −48.9, −81.5, 70.9, 56.0, −47.1−71.5, −44.4. While these samples have large errors they have little impact on the overall statistics.

**Figure 6 animals-13-00390-f006:**
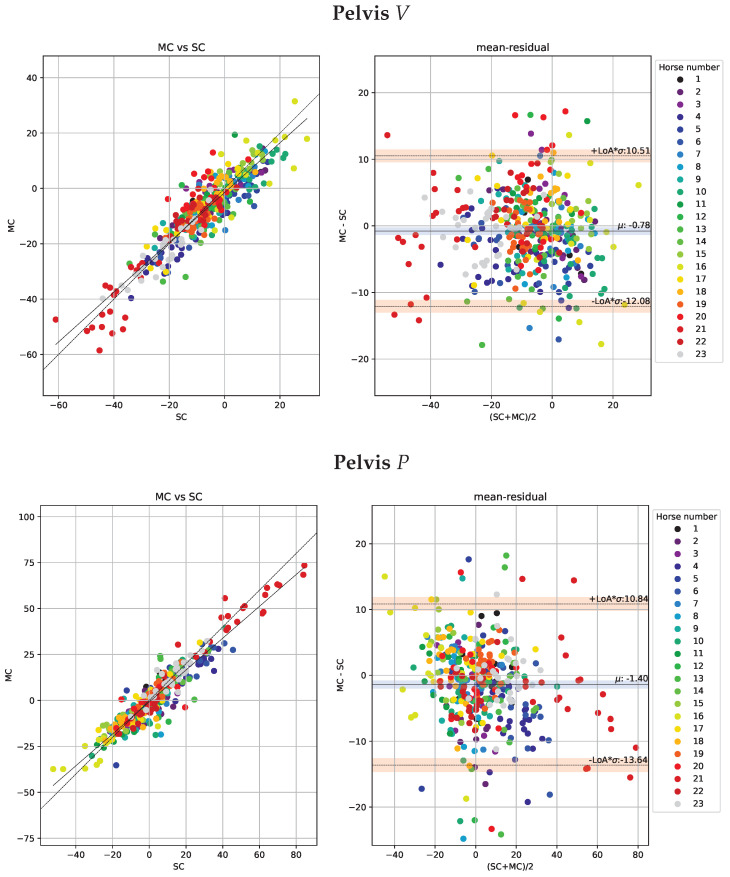
Scatter plots of the pelvic metrics obtained per stride (n = 404) observed by the multi-camera marker-based motion capture system (MC) plotted against the observation by the single-camera markerless system (SC) are shown in the left sub-panels. Agreements between the systems with limits of agreement (LoA) displayed as orange horizontal lines are presented in the Bland-Altman plots (right sub-panels). In the top row, we show the ΔV (MinDiff) and the bottom row we show ΔP (MaxDiff) defined in Section 2.7.1.

**Figure 7 animals-13-00390-f007:**
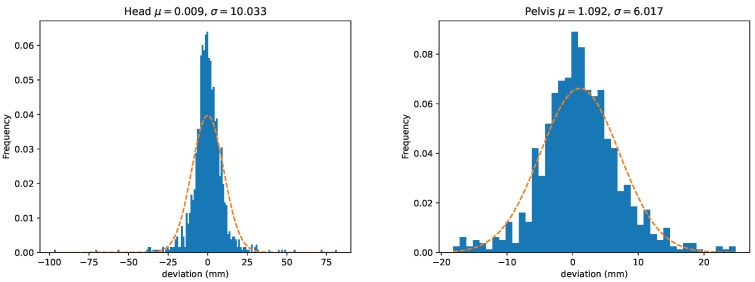
Distributions of the per-stride deviations for head (**left**) and pelvis (**right**). Here we have combined both peak and valley deviations defined in Section 2.7.1 (ΔV and ΔP). The dashed line displays the normal distribution estimated from the means and standard deviations of the between-systems deviations.

**Figure 8 animals-13-00390-f008:**
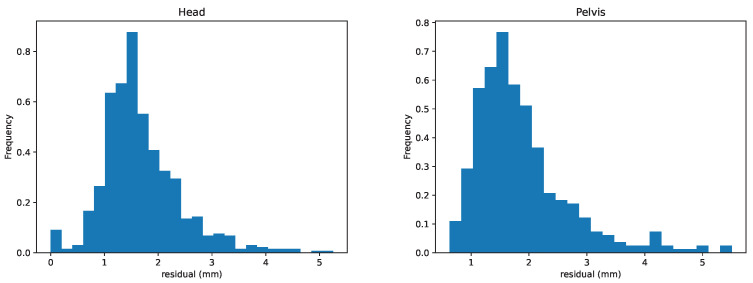
Distributions of the absolute values for stride mean residuals for head (**left**) and pelvis (**right**).

**Figure 9 animals-13-00390-f009:**
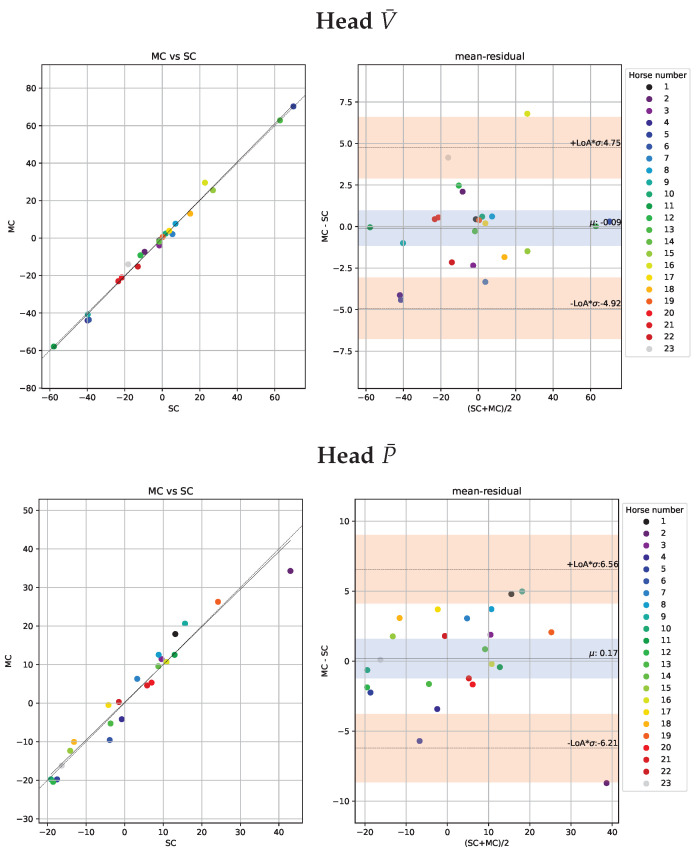
Trial-level scatter plots of the head metrics from the 23 included horses measured by the marker-based motion capture system (MC) versus the single-camera markerless system (SC) are presented in the left panel. Agreements between the systems, with limits of agreement (LoA) displayed as orange horizontal lines, are presented in the Bland-Altman plots in the right sub-panels. In the top row, we show $\Delta \bar{V}$ (trial mean MinDiff) and in the bottom row, we show $\Delta \bar{P}$ (trial mean MaxDiff) defined in Section 2.7.1.

**Figure 10 animals-13-00390-f010:**
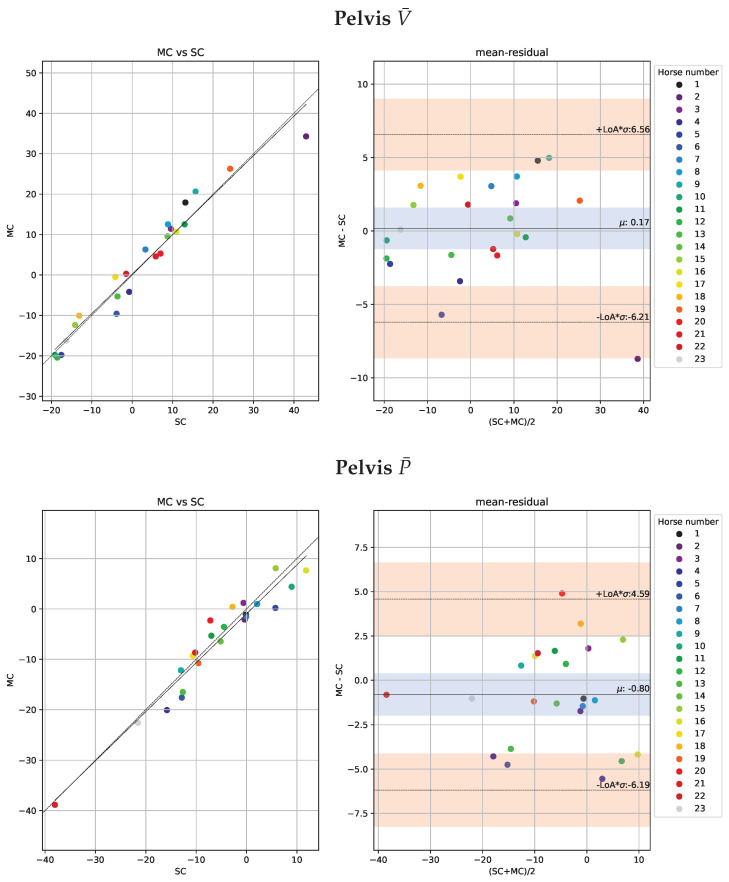
Trial-level scatter plots of the pelvic metrics from the 23 included horses measured by the marker-based motion capture system (MC) versus the single-camera markerless system (SC) are presented in the left panel. Agreements between the systems, with limits of agreement (LoA) displayed as orange horizontal lines, are presented in the Bland-Altman plots in the right sub-panels.In the top, we show $\Delta \bar{V}$ (trial mean MinDiff) and in the bottom row, we show $\Delta \bar{P}$ (trial mean MaxDiff) defined in Section 2.7.1.

**Figure 11 animals-13-00390-f011:**
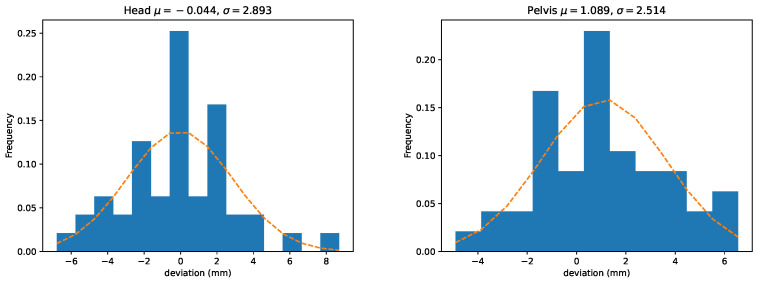
Distributions of the per-trial deviations for head (**left**) and pelvis (**right**). Here we have combined both peak and valley deviations defined in Section 2.7.1 ($\Delta \bar{V}$ and $\Delta \bar{P}$). The dashed line shows the normal distribution estimated from the means and standard deviations of the deviations.

**Table 1 animals-13-00390-t001:** Descriptive statistics for head measurements from the 23 included horses showing number of matched strides per trial (N) and the mean trial deviations between the two systems for the valley values $\Delta \bar{V}$ (MinDiff) and peak values $\Delta \bar{P}$ (MaxDiff). Also, the actual trial means for the $\bar{V}$ (MinDiff) and the $\bar{P}$ are presented per trial for the multi-camera marker-based (mc) and the single-camera markerless (sc) systems, followed by their within trial standard deviation (σV and σP). From the mc, the trial mean range of motion of the vertical displacement signal is presented.

Horse	N	$\Delta \bar{V}$	$\Delta \bar{P}$	${\bar{V}}^{\mathrm{sc}}$	${\bar{V}}^{\mathrm{mc}}$	${\bar{P}}^{\mathrm{sc}}$	${\bar{P}}^{\mathrm{mc}}$	$\sigma V^{\mathrm{sc}}$	$\sigma V^{\mathrm{mc}}$	$\sigma P^{\mathrm{sc}}$	$\sigma P^{\mathrm{mc}}$	${\bar{R}}_{\mathrm{mc}}$
1	16	−0.4	−4.8	−1.5	−1.1	13.1	17.9	15.2	15.4	18.1	16.5	66.6
2	23	−2.1	8.7	−9.4	−7.3	43.0	34.3	12.3	10.4	19.7	20.8	82.2
3	29	2.3	−1.9	−1.6	−3.9	9.5	11.4	17.7	15.0	13.6	14.1	70.4
4	38	4.1	3.4	−39.8	−44.0	−0.8	−4.2	14.3	15.7	14.8	16.8	71.2
5	28	−0.3	2.2	70.0	70.3	−17.5	−19.8	17.4	15.2	17.2	14.5	109.6
6	26	4.4	5.7	−39.2	−43.6	−3.9	−9.6	16.8	19.1	15.8	15.8	68.5
7	19	3.3	−3.1	5.4	2.1	3.3	6.3	13.5	14.2	19.9	21.1	77.5
8	22	−0.6	−3.7	7.1	7.7	8.8	12.5	15.7	11.9	11.0	10.4	73.6
9	29	1.0	−5.0	−39.7	−40.7	15.7	20.6	9.1	10.6	9.5	10.0	70.8
10	36	−0.6	0.6	1.8	2.4	−19.1	−19.8	22.3	22.0	20.7	21.4	95.2
11	27	0.0	0.4	−57.8	−57.9	13.0	12.5	11.5	13.1	16.4	15.2	90.1
12	28	−2.5	1.9	−11.7	−9.2	−18.5	−20.4	14.3	13.5	17.5	14.9	71.0
13	22	−0.0	1.6	62.8	62.8	−3.7	−5.3	8.9	8.2	10.8	10.3	79.8
14	22	0.3	−0.9	−1.6	−1.9	8.7	9.6	9.7	13.8	6.1	10.6	39.0
15	29	1.5	−1.8	27.0	25.5	−14.1	−12.4	13.8	13.1	11.2	11.6	75.2
16	19	−6.8	0.2	22.8	29.6	10.9	10.7	19.1	26.2	18.6	28.1	75.5
17	34	−0.2	−3.7	3.7	3.9	−4.2	−0.5	18.9	17.5	19.6	18.7	95.0
18	24	1.8	−3.1	14.9	13.1	−13.1	−10.1	8.3	7.9	13.6	11.4	57.9
19	35	−0.4	−2.1	0.2	0.6	24.2	26.3	6.9	6.4	8.6	6.0	51.0
20	41	−0.5	1.7	−21.6	−21.0	7.0	5.3	11.9	12.6	9.1	8.5	71.4
21	16	−0.4	1.2	−23.5	−23.0	5.8	4.6	8.4	7.7	6.7	7.4	43.0
22	36	2.2	−1.8	−13.0	−15.2	−1.5	0.3	8.5	8.2	12.3	10.5	50.9
23	56	−4.2	−0.1	−18.2	−14.0	−16.3	−16.2	18.7	17.6	18.8	18.9	74.7
mean	28.5	1.7	2.6	21.5	21.8	12.0	12.6	13.6	13.7	14.3	14.5	72.2

**Table 2 animals-13-00390-t002:** Descriptive statistics for pelvic measurements from the 23 included horses showing the number of matched strides per trial (N) and the mean trial deviations between the two systems for the valley values $\Delta \bar{V}$ (MinDiff) and peak values $\Delta \bar{P}$ (MaxDiff). Also, the actual trial means for the $\bar{V}$ (MinDiff) and the $\bar{P}$ are presented per trial for the multi camera marker based (mc) and the single camera markerless system (sc), followed by their within trial standard deviation (σV and σP). From the mc, the trial mean range of motion of the vertical displacement signal is presented.

Horse	N	$\Delta \bar{V}$	$\Delta \bar{P}$	${\bar{V}}^{\mathrm{sc}}$	${\bar{V}}^{\mathrm{mc}}$	${\bar{P}}^{\mathrm{sc}}$	${\bar{P}}^{\mathrm{mc}}$	$\sigma V^{\mathrm{sc}}$	$\sigma V^{\mathrm{mc}}$	$\sigma P^{\mathrm{sc}}$	$\sigma P^{\mathrm{mc}}$	${\bar{R}}_{\mathrm{mc}}$
1	13	1.0	−0.1	−0.1	−1.1	6.6	6.6	8.4	6.1	8.2	8.3	83.4
2	15	1.7	4.3	−0.4	−2.1	0.6	−3.8	11.0	8.6	10.0	8.6	82.3
3	16	−1.8	2.1	−0.6	1.2	−2.7	−4.8	6.5	4.0	8.9	5.6	82.6
4	24	4.3	6.5	−15.8	−20.1	21.0	14.5	9.3	7.9	9.6	8.1	92.5
5	17	5.5	−0.5	5.7	0.2	4.3	4.8	5.1	4.3	10.3	11.3	79.3
6	15	4.8	5.9	−12.8	−17.6	27.2	21.3	11.0	7.5	8.9	6.3	92.8
7	17	1.5	0.0	−0.1	−1.5	−5.2	−5.2	6.8	6.9	10.5	8.5	74.8
8	13	1.1	3.1	2.1	1.0	−8.6	−11.6	8.0	8.1	9.2	4.7	76.7
9	15	−0.8	2.2	−13.0	−12.2	−4.3	−6.5	7.1	4.6	6.0	6.0	67.5
10	22	4.6	2.1	9.0	4.4	2.2	0.1	6.5	7.7	15.8	16.0	77.7
11	15	−1.7	0.7	−7.0	−5.3	−11.9	−12.6	6.4	8.3	10.2	9.6	88.0
12	14	−0.9	1.1	−4.5	−3.6	2.6	1.5	6.4	4.8	7.3	7.9	64.7
13	11	3.9	2.4	−12.6	−16.5	9.0	6.6	9.7	6.9	8.1	8.5	70.8
14	14	1.3	0.9	−5.1	−6.4	13.0	12.0	5.8	6.7	9.0	8.6	74.8
15	14	−2.3	−3.0	5.8	8.1	−16.9	−13.9	5.7	3.1	6.0	3.3	75.3
16	13	4.2	−1.1	11.8	7.7	−25.8	−24.7	10.5	13.0	13.8	8.7	75.7
17	18	−1.4	−0.8	−10.6	−9.3	−1.4	−0.6	12.2	11.2	11.4	11.5	103.7
18	15	−3.2	−0.3	−2.8	0.4	−13.1	−12.8	5.6	4.6	7.1	3.3	85.5
19	26	1.2	−1.2	−9.6	−10.8	4.4	5.6	3.6	3.4	4.3	4.6	39.6
20	21	−4.9	1.3	−7.2	−2.3	0.1	−1.2	6.5	6.6	11.6	9.0	87.8
21	21	0.8	2.6	−38.0	−38.9	48.5	46.0	7.1	6.8	12.7	8.6	97.7
22	25	−1.5	2.8	−10.2	−8.7	−1.4	−4.2	6.6	5.1	5.5	6.1	67.0
23	30	1.0	0.4	−21.5	−22.6	12.5	12.1	8.9	9.8	8.8	8.0	72.6
mean	17.6	2.4	2.0	9.0	8.8	10.6	10.1	7.6	6.8	9.3	7.9	78.8

**Table 3 animals-13-00390-t003:** Results summary for the measurement deviations between the systems over the entire dataset. In the top 6 rows we provide combined per-trial deviations ($\bar{D}$) using both deviations for the normalised MinDiff ($\Delta \bar{V}$) and MaxDiff ($\Delta \bar{P}$). For both head and pelvis, we compute the overall absolute mean, maximum and minimum deviations. In the bottom two rows we provide mean root mean square deviations (RMSD) as a comparison of the shape of the vertical displacement signals.

Per Trial	
head $\bar{D}$	2.2 mm
pelvis $\bar{D}$	2.2 mm
head maxD	8.7 mm
pelvis maxD	6.5 mm
head minD	0.0 mm
pelvis minD	0.0 mm
**Per Stride**	
head mean RMSD	5.0 mm
pelvis mean RMSD	3.5 mm

## Data Availability

Raw stride-level data from all included horses (and all strides) are presented as Appendix A to the manuscript.

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
