# Peer review of "Is Markerless More or Less? Comparing a Smartphone Computer Vision Method for Equine Lameness Assessment to Multi-Camera Motion Capture"

_animals, 2023, doi:10.3390/ani13030390_

Round 1

Reviewer 1 Report

Excellent manuscript on an interesting and novel application, with extensive details on the comparison of the two methods. I have only one comment regarding the interpretation of the clinical relevance of the observed accuracy compared with the gold standard MC system. In lines 346-353 of the discussion, it is stated that earlier work by Keegan et al. revealed confidence intervals of 6mm and 3mm for head and pelvic asymmetries, respectively. Subsequently, the authors mention that these confidence intervals likely should be multiplied by a factor of two. I believe a bit more explanation on this topic would be beneficial to enhance clarity and facilitate interpretation. In the study by Pfau et al. (2016), 'reference' values of 8mm and 4mm have been reported, which is not exactly a factor of two, which may cause confusion. Please comment and elaborate on this point.

Author Response

Dear reviewer, 

Thank you very much for your comments and for dedicating time to improve our scientific work. 

The point-by-point answer can be found below. 

Kind regards, 

Elin Hernlund on behalf of the authors.

-----------------------------------------------------------

Excellent manuscript on an interesting and novel application, with extensive details on the comparison of the two methods.

Thank you!

I have only one comment regarding the interpretation of the clinical relevance of the observed accuracy compared with the gold standard MC system. In lines 346-353 of the discussion, it is stated that earlier work by Keegan et al. revealed confidence intervals of 6mm and 3mm for head and pelvic asymmetries, respectively. Subsequently, the authors mention that these confidence intervals likely should be multiplied by a factor of two. I believe a bit more explanation on this topic would be beneficial to enhance clarity and facilitate interpretation. In the study by Pfau et al. (2016), 'reference' values of 8mm and 4mm have been reported, which is not exactly a factor of two, which may cause confusion. Please comment and elaborate on this point.

We agreed that this could’ve been worded more clearly. The text has been changed to more explicitly describe the findings in the systems comparison study. It has also been written out what the suggested adjusted ranges are, and why we think this adjustment is also applicable to optical motion capture values. 

Reviewer 2 Report

Review - Animals 2076414

Overall, a very interesting body of work that seems to have great promise in the analysis of equine locomotion.

Line 1 – Need to define ‘computer vision’ as this is a novel term not readily understood in veterinary medicine

Line 27 – Replace ‘bottom’ with valley or related terms minima and maxima.

Line 43- Replace ’size’ with magnitude

Line 44 – Need to clarify what is meant by ‘numbers’

Line 45 – Inherent gait patterns is likely not the appropriate phrase as all gait patterns are inherent to the organism (versus external).  I think what you are referring to is laterality or subclinical asymmetries or altered movement patterns that are not judged to be related to lameness.  May also need to define lameness.

Line 62 – All of these ‘deep’ terms need defined or explained why they are all so deep.  This sounds very much like the proverbial ‘black box’ that many may disregard or question the validity.

Line 66 – It is not clear how ‘deep’ development allows quantitative motion capture from a single camera source.

Line 76 – Need a brief explanation of what a neural network is or does as it seems like it has to do more with brain function and not intuitively related to computer programming.

Line 110 – Markers placed on the dorsum of the head and pelvis

Figure 2 – Difficult to visualize marker placement in the photographs provided or description in the text.

Line 148 – While ‘push-off’ is a often used term, propulsive phase is more appropriate.

Line 159 – I am not aware of any of studies using inertial sensor or optical capture systems that normalize to ‘horse size’.  Is this height or body mass?

Section 2.3.1 – The present tense is used throughout, which is different from the rest of the manuscript where past tense is used.

Line 208 – How were the hoof movements tracked if no reflective markers were used?  Is this only for the photographic method?

All the reported formulas are beyond my understanding so I cannot comment on the validity of their application or use.

Section 3.1 – It is unusual to state that data will be reported in a figure (‘we provide’, ‘we present’), rather than just stating results (Figure X).

Line 295 – is this a complete sentence – as,…or does it refer to the figures?

Figure 4 – it takes some time to figure out the numerous images and hard to see due to small size.  But after a bit it is understandable.

Figure 5 and 6 – Not able to read any of the axis labels. It is not intuitive what B and P refer to – recommend using minima and maxima.

Figure 7, 8, 11 – Switch the graphs so head is on the left and pelvis on the right.

Page 11 – Formatting error

Figure 9 and 10 – Completely unreadable figures.

Table 3 and 4 – Header text needs to be enlarged.

Line 311 –Was there an actual lack of light or not as a limitation?

Line 317 – Need to define lens optics and image resolution.  Mentioning iPhone 12 Pro Max does not provide any useful details regarding camera settings.

How sensitive is the SC-system to obliquity?  Does the iPhone need to be placed directly behind the horse, or can it be placed off to one side or the other?

Line 321 – Seems that protraction-retraction is best measured for the lateral view.  What are the possibilities for using this system from a different perspective?

Line 323 – Does this refer to “deeper” deep learning?  Sorry a little humor.

Line 328 – “previously recorded between-measurement-variation” – from where and for what?

Line 340 – ‘pelvic’ and ‘magnitude’

Line 343 – why capitalize ‘computer vision’?  See prior comment on defining what this is as it is not intuitive that computers ‘see’.

Line 344 – Reword or move sentence as it is out of place here and no context is provided.

Line 349 – When is a ‘millimeter scale’ not a millimeter scale?  This defiles logic.

Line 354 – This section needs to be moved to the limitations section as it relates to observed and not theoretical limitations.

Line 362 – It is always off-putting to read that ‘this is the first study’.  If novel data (i.e., first) was not actually presented, then it is not an original research article, but rather a review or editorial.

Line 369 – Just because the method has been used in humans does not give if validity.  Provide some context here – was it valid, clinically useful, patients examined (neurologic, lame) etc.

Line 372 – Two more items to add to the limitations section

Line 383 – Define terms ‘machine learning’ and ‘big-data research’ as it relates to gait analysis and prior terms used in this work – ‘deep learning’ and ‘neural networks’.  These are vague terms and seem to be catch-phrases that are not readily understood by the veterinary readership.

Line 389 – ‘clearly lower than observed levels of between “run” (replace with trial) variation’ – observed form where?  Your work or others?

Line 390 – ‘better clinical precision’ compared to what?  How does ‘precision’ relate to longitudinal observations, defining ‘lameness’ thresholds or characteristics, or improving clinical decision-making skills?

Author Response

Dear reviewer,

Thank you for all your valuble comments: They have helped improve the manuscript substantially.

Please find the point-by-point answers below. 

Overall, a very interesting body of work that seems to have great promise in the analysis of equine locomotion.

Thank you!

Line 1 – Need to define ‘computer vision’ as this is a novel term not readily understood in veterinary medicine

Thank you. The term has been clarified. 

Line 27 – Replace ‘bottom’ with valley or related terms minima and maxima.

We have replaced bottom with valley.

Line 43- Replace ’size’ with magnitude

Thank you, it now reads magnitude. 

Line 44 – Need to clarify what is meant by ‘numbers’

Changed to “asymmetry levels”.

Line 45 – Inherent gait patterns is likely not the appropriate phrase as all gait patterns are inherent to the organism (versus external).  I think what you are referring to is laterality or subclinical asymmetries or altered movement patterns that are not judged to be related to lameness.  May also need to define lameness.

Yes, thank you. We have changed to “laterality”.

Line 62 – All of these ‘deep’ terms need defined or explained why they are all so deep.  This sounds very much like the proverbial ‘black box’ that many may disregard or question the validity.

We have clarified that the term “deep” refers to the layered hierarchy of functions (“neurons”) in a neural network. 

Line 66 – It is not clear how ‘deep’ development allows quantitative motion capture from a single camera source.

We have added the development of human pose estimation in order to make this clearer.

Line 76 – Need a brief explanation of what a neural network is or does as it seems like it has to do more with brain function and not intuitively related to computer programming.

We agree that it can be made clearer that neural networks in this context pertain to computer algorithms and not biological brains. As such we have made a clarification on line 74. Hopefully this clears up any confusion early on in the description. More in depth details of what a neural network is, is thoroughly described in the cited literature [24-26]. We don’t believe that delving any deeper into this topic is within scope for this manuscript. 

Line 110 – Markers placed on the dorsum of the head and pelvis

The anatomical placement of the markers has been clarified in the text. 

Figure 2 – Difficult to visualize marker placement in the photographs provided or description in the text.

The figure legend has been altered to clarify the marker placement.

Line 148 – While ‘push-off’ is a often used term, propulsive phase is more appropriate.

A much appreciated suggestion! Propulsive phase is now used.  

Line 159 – I am not aware of any of studies using inertial sensor or optical capture systems that normalize to ‘horse size’.  Is this height or body mass?

The most commonly used tool for objective lameness, the Lameness Locator, performs a normalization process where the asymmetry metrics are related to the range of motion of the symmetrical component H2 of the signal. 

Section 2.3.1 – The present tense is used throughout, which is different from the rest of the manuscript where past tense is used.

The section has been modified and all text should now be in past tense

Line 208 – How were the hoof movements tracked if no reflective markers were used?  Is this only for the photographic method?

Yes, this has now been clarified in the materials and methods section. 

All the reported formulas are beyond my understanding so I cannot comment on the validity of their application or use.

Section 3.1 – It is unusual to state that data will be reported in a figure (‘we provide’, ‘we present’), rather than just stating results (Figure X).

We have rewritten the section according to the suggestion. 

Line 295 – is this a complete sentence – as,…or does it refer to the figures?

We are sorry about this formatting error. The sentence is continued by an equation, but the figures had been mistakenly placed inbetween. 

Figure 4 – it takes some time to figure out the numerous images and hard to see due to small size.  But after a bit it is understandable.

We have put the two subpanels in a column to enable increased size for improved readability.

Figure 5 and 6 – Not able to read any of the axis labels. It is not intuitive what B and P refer to – recommend using minima and maxima.

The axis labels are improved and the wording and labelling reworked. 

Figure 7, 8, 11 – Switch the graphs so head is on the left and pelvis on the right.

The figures have been reordered as requested. 

Page 11 – Formatting error

Corrected

Figure 9 and 10 – Completely unreadable figures.

The figure format has been adjusted to improve readability

Table 3 and 4 – Header text needs to be enlarged.

Correction made. 

Line 311 –Was there an actual lack of light or not as a limitation?

You can now find the limitations section as a subheading under the discussion (as suggested by the academic editor). We have clarified that we did not test under low lighting conditions within this study. In general, outdoor daylight provides much more light than the indoor environment used here.  

Line 317 – Need to define lens optics and image resolution.  Mentioning iPhone 12 Pro Max does not provide any useful details regarding camera settings. How sensitive is the SC-system to obliquity?  Does the iPhone need to be placed directly behind the horse, or can it be placed off to one side or the other?

We agree that the use of camera optics is not well defined. We changed the wording to be more specific and well defined within the context of the manuscript. 

Investigating obliquity was out of scope for the comparison, but is certainly something that is interesting to look into in future works. Since we only measure vertical displacements, we observe that the SC system is not sensitive to small changes in camera position as long as the pelvis and head is clearly visible from the camera.

Line 321 – Seems that protraction-retraction is best measured for the lateral view.  What are the possibilities for using this system from a different perspective?

We agree. This is certainly true for 2D extraction of motion parameters from the image. 3D lifting could potentially allow retraction and protraction to be extracted independent of view point, given that the “horse body model” would be trained on large and precise dataset. This has been done for human bodies, see: Wang, J., et al. (2021). Deep 3D human pose estimation: A review. Computer Vision and Image Understanding, 210. https://doi.org/10.1016/j.cviu.2021.103225 

Line 323 – Does this refer to “deeper” deep learning?  Sorry a little humor.

Much appreciated:)

Line 328 – “previously recorded between-measurement-variation” – from where and for what?

We have clarified the origin of the referenced data in Hardeman et al 2019. 

Line 340 – ‘pelvic’ and ‘magnitude’

Thank you. The corrections are made. 

Line 343 – why capitalize ‘computer vision’?  See prior comment on defining what this is as it is not intuitive that computers ‘see’.

Capital letters have been removed. 

Line 344 – Reword or move sentence as it is out of place here and no context is provided.

The sentence has been reworded to clarify that it refers to the study discussed in this section. 

Line 349 – When is a ‘millimeter scale’ not a millimeter scale?  This defiles logic.

We agree this does not appear logical. The output of this IMU-system is presented in mm values but are in fact not corresponding to “true” millimetres. As we understand their method this has to do with scaling and normalisation. We have altered the section to make it less confusing. 

Line 354 – This section needs to be moved to the limitations section as it relates to observed and not theoretical limitations.

We would love to keep the section under discussion since we do feel it is a discussion of our results. We have however added more text about challenging recording conditions to the limitations section. We hope this is acceptable. 

Line 362 – It is always off-putting to read that ‘this is the first study’.  If novel data (i.e., first) was not actually presented, then it is not an original research article, but rather a review or editorial.

We see your point. The sentence has been omitted.

Line 369 – Just because the method has been used in humans does not give if validity.  Provide some context here – was it valid, clinically useful, patients examined (neurologic, lame) etc.

We have added a sentence to clarify “The level of precision described is said to open up for several potential applications in human medicine.”

Line 372 – Two more items to add to the limitations section. 

We feel again that these are not limitations of the study but rather a discussion of the usefulness of the technique related to our results. We hope the reviewer can agree to this.  

Line 383 – Define terms ‘machine learning’ and ‘big-data research’ as it relates to gait analysis and prior terms used in this work – ‘deep learning’ and ‘neural networks’.  These are vague terms and seem to be catch-phrases that are not readily understood by the veterinary readership.

Thank you for drawing our attention to this. We have now added a short description of machine learning and omitted “dig data”. 

Line 389 – ‘clearly lower than observed levels of between “run” (replace with trial) variation’ – observed form where?  Your work or others?

“Run” is replaced by “trial” and a reference is added. 

Line 390 – ‘better clinical precision’ compared to what?  How does ‘precision’ relate to longitudinal observations, defining ‘lameness’ thresholds or characteristics, or improving clinical decision-making skills?

We agree that the sentence was confusing. It has nor been rewritten: The ease of use of the system makes repeated observations of a horse's lameness more doable, which can provide more objective data points for better treatment evaluation.

Reviewer 3 Report

This paper must be reviewed by someone with knowledge of advanced biomechanics & statistics

I cannot comment on the appropriateness of the assumptions that were made when processing the data for analysis. Many data transformations were made.

The paper describes a comparison of optical motion capture and a computed assessment of movement symmetry determined by smart phone video analysis of unmarked horses. The study was small comprising only 25 horses, two of which were removed from analysis

There is no section in the Materials and Methods describing what statistical analyses were performed.

There is no lay person's Summary.

Line 14  & elsewhere   Is it really biologically meaningful to be quoting  data to 2 decimal places of a mm?

Line 21 a horse not 'the'.    This type of grammatical error is commonplace throughout the manuscript. e.g., Line 53 a horse's

Line 46 unveiling is not an appropriate word in this context

Further investigation

Response to diagnostic anaesthesia and assessment of behavioural indicators of pain may assist in differentiating which gait asymmetries are pain induced.

Line 68 What is meant by 'an incipient development '?

Line 101 Was required would be more accurate than could be sought

Line 103 Was speed standardised?

Was the handler always on the same side?

Line 109 – marker placement. Where precisely were the markers placed?

How many markers were used?

How were the markers attached?

Did the same person place the markers on all horses?

Line 325 You have only assessed in hand at trot

Line 365 Please check reference numbers throughout the manuscript

Should this be 29?

Line 366 was similar (not were)

Line 379  This study also had a low sample size! Was a sample size calculation performed?

Line 335

‘….relevant perspective question of precision when it comes to gait analysis, in terms of helping

the clinician of understanding the horses orthopaedic issues, should maybe be directed to

the precision in seeing the horse for a momentary visit.’

Not clear what this means

Line 357 interpretation

Line 361 How would the system cope with a horse with a naturally high tail carriage (e.g., many Arabians & some Warmbloods) which partially obscures the pelvis

How would the system cope with piebald or skewbald coloured horse?

How reliable will the system be in rain? – or filming directly into the sun?

Author Response

Dear Reviewer, 

Thank you for your many relevant comments. Working with them has improved the manuscript and the readability to our veterinary audience. 

Please find the point-by-point answers below.

The paper describes a comparison of optical motion capture and a computed assessment of movement symmetry determined by smart phone video analysis of unmarked horses. The study was small comprising only 25 horses, two of which were removed from analysis

There is no section in the Materials and Methods describing what statistical analyses were performed.

Thank you for pointing this out. We have now added a section under the heading “System comparison” called “statistical analysis”. A very important improvement of the manuscript. 

There is no lay person's Summary.

A simple summary is now added.  

Line 14  & elsewhere   Is it really biologically meaningful to be quoting  data to 2 decimal places of a mm?

Fair point, we reduced to one decimal for mm values.

Line 21 a horse not 'the'.    This type of grammatical error is commonplace throughout the manuscript. e.g., Line 53 a horse's

Thank you. “The” is replaced with “a” in lines 23 and 53. 

Line 46 unveiling is not an appropriate word in this context. Further investigation

We have made the requested change. 

Response to diagnostic anaesthesia and assessment of behavioural indicators of pain may assist in differentiating which gait asymmetries are pain induced.

 We do completely agree! And we do perform research on how pain behaviours associate with gait asymmetry in horses. Longitudinal monitoring of both gait and behavioural patterns would allow us to understand much more. 

Line 68 What is meant by 'an incipient development '?

 We aimed to describe that this development has just come into existence.

Line 101 Was required would be more accurate than could be sought

 Thank you. “Could be sought” is changed to “was required”.

Line 103 Was speed standardised? Was the handler always on the same side?

No, horses were running at the preferred speed of the handler. The handler was always running on the left side of the horse. This has been clarified in the materials and methods section. 

Line 109 – marker placement. Where precisely were the markers placed? How many markers were used? How were the markers attached? Did the same person place the markers on all horses?

We have now better described the placement of the markers in the text. Only two markers were used in this study, but in our teaching hospital we routinely place 11 markers on each horse undergoing objective lameness assessment during orthopaedic work up. Only two markers are used for the clinical assessment, the rest are recorded for potential future research purposes. Trained staff placed the markers. 

Line 325 You have only assessed in hand at trot

We have added “when trotted in hand on the straight” to the sentence. 

Line 365 Please check reference numbers throughout the manuscript. Should this be 29?

Thank you for noticing! We had misplaced the intended reference  (Wang 2021) with another (Wang 2020).

Line 366 was similar (not were)

Correction made.

Line 379  This study also had a low sample size! Was a sample size calculation performed?

We agree that it would have been ideal to perform a pre-study sample size calculation. This was difficult since the expected deviation between the systems was not known. Since the stride level comparison was the most relevant, we were quite confident that we would reach enough strides.  

Line 335

‘….relevant perspective question of precision when it comes to gait analysis, in terms of helping

the clinician of understanding the horses orthopaedic issues, should maybe be directed to

the precision in seeing the horse for a momentary visit.’ Not clear what this means

 Sorry about this confusing sentence. We have the clinical experience from using objective gait analysis that measurements are often very precise. But one questions if the observed asymmetry was there the day, or the week before.  We have tried to explain this better. 

Line 357 interpretation

Thank you, corrected.

Line 361 How would the system cope with a horse with a naturally high tail carriage (e.g., many Arabians & some Warmbloods) which partially obscures the pelvis.

You are right, as stated in the discussion a high tail carriage can sometimes extensively obscure the pelvis and thus be a hindrance to the extraction of data from the SC-system.

 How would the system cope with piebald or skewbald coloured horse?

Horse colour in not an issue for the SC system. 

How reliable will the system be in rain? – or filming directly into the sun?

Thanks for this relevant question. We have added a statement about this to the limitations section.

Round 2

Reviewer 3 Report

The revised version of this manuscript has addressed the majority of my previous comments, but there are a few other new comments. Throughout the manuscript the term peaks and valleys has been used, whereas in this context I believe that peaks and troughs would be preferable. Not all the figure legends could be read independently from the text.

Line 50 However would be a better word than but

Line 70 time-consuming may be better than time demanding

Line 81 ‘a progressive movement towards’ would be more readily understood than an incipient development

Line 86 a clinician (not the)

Line 87 However would be a better word than but

M&M Please clarify if horses were lame on a single limb or exhibited lameness in more than 1 limb

Please discuss the use of the SC system for assessment of horses with lameness in > 1 limb.

It is implied in the Discussion that you have validated the SC system for all lameness evaluation but that is not necessarily the case. There are horses with no measurable asymmetries in hand which are lame when ridden, sometimes only in specialised conditions, such as 10m diameter circles.

Figure 2 legend The markers on the poll and between the tubera sacrale were used

Line 120 do you mean that the cameras were placed approximately 4m apart?

Line 154   were available would be better than were found

Line 157 replace 'due to that' with because

Lines 321  It seems a shame that so much reliance is placed upon reference 21 which investigated a small number of so-called 'owner-sound' horses (a dreadful term that should be eliminated from the scientific literature). It seems highly likely that at least some of these horses would have been lame, probably on >1 limb, which would account for the variabilities seen.

I don't see the value of the statement 'It is often difficult for the clinician to understand if an observed low degree asymmetry is  a reflection of a horse’s persistent motion pattern or associated to the owners complaint. Thus, repeated observations of motion asymmetry can add to the precision of the diagnostic procedure.'

How can measurement (as performed by AI) differentiate between what is the owner's complaint and  what is observed?? The measurement can verify asymmetry - but that asymmetry assessed in hand may still not reflect the owner's complaint when the horse is ridden.

Limitations: small number of horses – you cannot possibly have evaluated the wide spectrum of lameness types that  can be encountered in clinical practice.

I rely on whomever else is reviewing for verification of some of the analyses performed which are beyond my statistical knowledge.

Please check carefully for typographical errors, sapcing etc.  e.g., MinDiff and MaxDiffand

Author Response

Dear reviewer, 

Many thanks for your additional comments. They have improved the readability of the manuscript substantially. We provide answers to your specific comments below. 

Throughout the manuscript the term peaks and valleys has been used, whereas in this context I believe that peaks and troughs would be preferable. 

We agree that through is a good term, but the term valley was requested by another reviewer. Therefore we made this alteration of the revised manuscript. We have searched scientific papers and found valleys to be commonly used elsewhere in biomechanical literature (e.g. https://www.researchgate.net/publication/287210217_Application_of_Hybrid_Multi-Resolution_Wavelet_Decomposition_Method_in_Detecting_Human_Walking_Gait_Events). We also explain the valleys to be local minima. We hope it is ok to stick to valley according to Reviewer 2’s earlier suggestion. 

Not all the figure legends could be read independently from the text.

Thank you for pointing this out. We have revised the legends to figure 4, 5, 6, 9, 10 and table 1 to make them stand alone. Also the supplementary figure’s legends have been updated. 

Line 50 However would be a better word than but

Thank you. The suggested change has been made. 

Line 70 time-consuming may be better than time demanding

Thank you. The alteration has been made. 

Line 81 ‘a progressive movement towards’ would be more readily understood than an incipient development

We agree. Thank you for improving the readability. 

Line 86 a clinician (not the)

The requested change has been made. 

Line 87 However would be a better word than but

Thank you for the suggestion. ‘But’ is changed to ‘however’. 

M&M Please clarify if horses were lame on a single limb or exhibited lameness in more than 1 limb

Please discuss the use of the SC system for assessment of horses with lameness in > 1 limb.

Thank you for pointing this out. We have clarified in the introduction that the vertical asymmetry metrics for head and pelvis are sensitive indicators of single limb lameness (Line 49). Horses with bilateral lameness of the exact same magnitude on contralateral limbs would not show this type of asymmetry on the straight. Instead, they would be likely to reduce the impulse per stride by a relative decrease in stride length (higher relative stride frequency). For the aim of this study it was not important to exclude horses with bilateral lameness since we perform a precise stride-by-stride comparison between two measurement systems.

It is implied in the Discussion that you have validated the SC system for all lameness evaluation but that is not necessarily the case. There are horses with no measurable asymmetries in hand which are lame when ridden, sometimes only in specialised conditions, such as 10m diameter circles.

We acknowledge that horses can display signs of orthopaedic disease only in specialised conditions. This is however not within the scope of this study to investigate. The specific purposes of this study was to (stride-by-stride and trial-by-trial) compare the vertical asymmetry metrics of the included horses. These measurements are what the current lameness quantifying systems use and are therefore most relevant to study. We believe it is clear from the aims, results and discussion that we do not claim any other capability of the systems compared. 

Figure 2 legend The markers on the poll and between the tubera sacrale were used

Thank you, we have changed to plural. 

Line 120 do you mean that the cameras were placed approximately 4m apart?

We have now clarified that 4 m is height over the ground and not related to the field of view overlap.  

Line 154   were available would be better than were found

The alteration has been made. 

Line 157 replace 'due to that' with because

Thank you. The alteration has been made. 

Lines 321  It seems a shame that so much reliance is placed upon reference 21 which investigated a small number of so-called 'owner-sound' horses (a dreadful term that should be eliminated from the scientific literature). It seems highly likely that at least some of these horses would have been lame, probably on >1 limb, which would account for the variabilities seen.

Stride and trial variability and their relation to bilateral lameness is an interesting topic. To our knowledge the association between variability of vertical asymmetry metrics and bilateral lameness is not well studied. The reason for our emphasis on reference 21 is that it is the only study showing between trial variability with a state-of-the-art position measuring system. It therefore provides very important reference data for our specific comparison. 

Owner-sound is for sure a vague term when it comes to the clinical status of a horse's locomotor system. But we suggest it should be seen as a term describing the owner's opinion on the horse, not the actual presence of disease. And as such, we believe it can have an important place even in the scientific literature. 

I don't see the value of the statement 'It is often difficult for the clinician to understand if an observed low degree asymmetry is  a reflection of a horse’s persistent motion pattern or associated to the owners complaint. Thus, repeated observations of motion asymmetry can add to the precision of the diagnostic procedure.' How can measurement (as performed by AI) differentiate between what is the owner's complaint and  what is observed?? The measurement can verify asymmetry - but that asymmetry assessed in hand may still not reflect the owner's complaint when the horse is ridden.

The statement has been removed. 

Limitations: small number of horses – you cannot possibly have evaluated the wide spectrum of lameness types that can be encountered in clinical practice.

We have added the number of study subjects to the limitations section in the discussion. 

We would like to underline that we do not claim to classify or measure all types of lamenesses in this study. We only compare the vertical asymmetry metrics which are (so far) the most used and sensitive lameness metrics in the scientific literature on equine lameness. We have experience from both the clinic and biomechanical studies that other motion parameters are often interesting as well, and can be an important complement to the vertical asymmetry of the axial body segments. These are however not in the scope of the current study. 

I rely on whomever else is reviewing for verification of some of the analyses performed which are beyond my statistical knowledge. Please check carefully for typographical errors, sapcing etc.  e.g., MinDiff and MaxDiffand

Spaces have been added. Thank you.